# The Role of Fecal Microbiota Transplantation in the Allogeneic Stem Cell Transplant Setting

**DOI:** 10.3390/microorganisms11092182

**Published:** 2023-08-29

**Authors:** Elisabetta Metafuni, Luca Di Marino, Sabrina Giammarco, Silvia Bellesi, Maria Assunta Limongiello, Federica Sorà, Filippo Frioni, Roberto Maggi, Patrizia Chiusolo, Simona Sica

**Affiliations:** 1Dipartimento di Diagnostica per Immagini, Radioterapia Oncologica e Ematologia, Fondazione Policlinico Universitario Agostino Gemelli IRCCS, 00168 Rome, Italy; sabrina.giammarco@policlinicogemelli.it (S.G.); silvia.bellesi@policlinicogemelli.it (S.B.); mariaassunta.limongiello@guest.policlinicogemelli.it (M.A.L.); federica.sora@policlinicogemelli.it (F.S.); patrizia.chiusolo@unicatt.it (P.C.); simona.sica@unicatt.it (S.S.); 2Sezione di Ematologia, Dipartimento di Scienze Radiologiche ed Ematologiche, Università Cattolica del Sacro Cuore, 00168 Rome, Italy; luca.dimarinomd@gmail.com (L.D.M.); filippo.ffrioni@outlook.com (F.F.); robertomaggi20@gmail.com (R.M.)

**Keywords:** fecal microbiota transplantation, graft-versus-host disease, allogeneic hematopoietic stem cell transplantation, multi-drug resistant, clostridium difficile infection, probiotic, prebiotic, diversity

## Abstract

Microbiota changes during allogeneic hematopoietic stem cell transplantation has several known causes: conditioning chemotherapy and radiation, broad-spectrum antibiotic administration, modification in nutrition status and diet, and graft-versus-host disease. This article aims to review the current knowledge about the close link between microbiota and allogeneic stem cell transplantation setting. The PubMed search engine was used to perform this review. We analyzed data on microbiota dysbiosis related to the above-mentioned affecting factors. We also looked at treatments aimed at modifying gut dysbiosis and applications of fecal microbiota transplantation in the allogeneic stem cell transplant field, with particular interest in fecal microbiota transplantation for graft-versus-host disease (GvHD), multidrug-resistant and clostridium difficile infections, and microbiota restoration after chemotherapy and antibiotic therapy.

## 1. Literature Review

We performed a review through PubMed electronic database: https://pubmed.ncbi.nlm.nih.gov/ accessed on 15 of June 2023). We searched for the relevant articles published in the last decade up to June 2023. The following strings was used to perform the literature search: (stem cell transplant * OR HSCT * OR allogeneic stem cell transplant * OR hematopoietic stem cell transplant) AND (fecal microbiota transplant * OR FMT).

## 2. Introduction

The intestine is the main site of bacterial, viral, and fungal colonization. All these microorganisms do not merely act as commensal organisms, but actively participate in the digestion of complex carbohydrates and interact with the host immune system in a manner that we still do not fully understand. A healthy bacterial microbiota is involved in heterogeneous activities: development and maturity of the host immune system, digestion of food, synthesis of essential amino acids, short-chain fatty acids (SCFAs), and vitamins, regulation of the immune response, and enhancement of the resistance to pathogenic infection [1]. The vast majority of the bacteria belong to the *Bacteroidetes* and *Firmicutes phyla*; these bacteria are in equilibrium with the host’s innate immune system and help to maintain homeostasis, which directly affects host health when altered [2]. The gut microbiota of each individual contains many unique strains not found in others, and inter-individual differences in microbiota composition are much larger than intra-individual variations [3].

It has been known since the 1970s that the microbiota is involved in graft-versus-host disease (GvHD) pathogenesis, as demonstrated by the van Bekkum group in 1974, where mice raised in a germ-free environment did not develop gastrointestinal (GI) GvHD. Just recently, we have achieved a partial understanding of the panorama and the intricacies of the microbiota and GvHD [4,5,6]. The microbiota-derived signals can indirectly influence regulatory T lymphocytes (Tregs) by activating innate, gut-resident, antigen-presenting cells (APCs) known as CD103 + CD11b + dendritic cells (DCs) [7]. These DCs subsequently promote adaptive anti-inflammatory responses of Treg cells by producing molecules such as transforming growth factor β (TGF-β) and the vitamin A metabolite, retinoic acid. Tregs are not only responsible for maintaining immunological tolerance in tissues but also actively contribute to tissue repair through the production of amphiregulin [8]. This mechanism is important for homeostasis and modulates the immune response against the gut flora. An impaired mechanism can lead to a pro-inflammatory state that can be common in inflammatory bowel diseases, non-alcoholic steatohepatitis, type 2 diabetes, obesity, and, with some peculiarities, in acute GvHD (aGvHD) [9,10].

To recognize the complexity of the microbiota, it is essential to acknowledge that any factors disrupting or weakening this system can predispose to infections and autoimmunity. In the setting of allogeneic hematopoietic stem cell transplantation (HSCT), damage to the microbiota is not solely due to the conditioning regimen but also to the preceding chemotherapy and antibiotic treatments, which consistently undermine the microbiota [11,12].

This article aims to review the current knowledge about the close link between microbiota and HSCT. We have sought to highlight the impact of therapies on microbiota composition and how gut dysbiosis affects transplant outcomes. We also looked at treatments aimed at modifying gut dysbiosis and applications of fecal microbiota transplantation (FMT) in the HSCT setting. In Figure 1, we have represented major factors affecting α-diversity of fecal microbiota. Alpha-diversity is a variable that reflects the number of unique bacterial *taxa* present and their relative frequencies [4].

## 3. Microbiota Changes during Allogeneic Stem Cell Transplantation

Significant changes were reported in microbiota composition during the HSCT procedure, both in terms of diversity and in terms of taxonomy [13]. These changes might contribute to post-transplant outcomes, such as GvHD incidence, transplant-related mortality (TRM), infectious complications, and overall survival (OS). A reduced intestinal microbial diversity after HSCT was reported to affect survival outcomes [14], while a low microbial diversity at engraftment was significantly associated with a high risk of TRM and severely reduced OS [15,16]. Predominant *genera* in samples obtained from patients with reduced microbial diversity were *Enterococcus*, *Streptococcus*, *Enterobacteriaceae,* and *Lactobacillus* [4,15]. Patients who died showed an abundance of *Proteobacteria* including *Enterobacteriaceae*, while surviving patients showed an abundance of *Lachnospiraceae* and *Actinomycetaceae* [15].

### 3.1. Antibiotics Affect the Gut Microbiota

A patient candidate for HSCT has already received several courses of broad-spectrum antibiotics. These therapeutic interventions have profoundly influenced the composition of the patient’s gut microbiota [17]. Notably, the most significant impact on the microbiota arises from the antimicrobial regimens administered both prior to and after transplantation. Different antibiotic classes exert distinct effects on the gut microbiota, which largely depend on the activity spectrum of the drug and may promote a different proinflammatory pattern in the gut flora. Particularly, the use of broad-spectrum antibiotics leads to a reduction in microbiota diversity and richness.

It has been demonstrated in mice that treatment with levofloxacin and cefepime had selected a gut flora with high abundances of *Clostridia* compared with meropenem-treated mice, and this difference was consistent with fecal butyrate levels. Additionally, both levofloxacin- and cefepime-treated mice had significantly lower abundances of *Bacteroides thetaiotaomicron*, compared with meropenem-treated mice. Notably, meropenem usage has been associated with the proliferation of *Bacteroides thetaiotaomicron*, a Gram-negative obligate anaerobe, that exhibits the capacity to metabolize dietary polysaccharides and host-derived glycans, including mucin. Therefore, *Clostridia* regulates intestinal immunity through SCFA production, while *Bacteroides thetaiotaomicron* is detrimental in determining mucus integrity. This observation provides a plausible explanation for the correlation between the utilization of carbapenems, frequently employed in the context of pre-engraftment febrile neutropenia, and the heightened severity of GI GvHD [18].

Different antibiotics used in the HSCT setting might impact GvHD-related mortality, probably due to the modification induced by antibiotics in microbiota composition [4]. A study conducted by Shono et al. on 857 patients who underwent T-cell-replete HSCT showed that among the twelve most frequently used antibiotics, piperacillin-tazobactam and imipenem-cilastatin were associated with different GvHD-related mortality rates. Additionally, these antibiotics were linked to an increased incidence of grade II–IV aGvHD with a higher occurrence of upper GI GvHD. Conversely, exposure to aztreonam or cefepime correlated with reduced GvHD-related mortality. Microbial composition in subjects and mice treated with imipenem-cilastatin showed a decreased presence of *Clostridiales*, which is believed to regulate anti-inflammatory processes in the gastrointestinal tract, inducing Tregs via SCFAs metabolites [19]. The use of antibiotic prophylaxes like ciprofloxacin and broad-spectrum systemic antibiotics reduced commensal bacteria, favoring the overgrowth of *Enterococci* [20]. Antibiotic therapy is not only used for infections but is also employed as a gut decontamination strategy. Weber et al. compared two decontamination schedules: ciprofloxacin 500 mg twice a day and metronidazole 500 mg three times a day, starting 8 days before HSCT until 14 days post-engraftment (n = 200) and rifaximin 200 mg twice a day (n = 194). Results indicated that gut decontamination with rifaximin was associated with similar rates of infectious complications compared with ciprofloxacin/metronidazole but preserved a high intestinal microbiota diversity and mitigated the negative effects of systemic antibiotics on microbial composition [21]. Previously hospitalized patients compared with de novo admitted exhibited a reduced expression of predominant commensal strains together with an increased presence of *Enterococci* [20]. The selection of antibiotics with a narrower spectrum of activity, particularly targeting anaerobic bacteria, may mitigate intestinal GVHD by minimizing the extent of microbiota alterations [22]. The results are presented Table 1.

### 3.2. Conditioning Affects the Gut Microbiota

By the time a patient arrives at the time of transplantation, he/she has already received varying amounts of chemotherapy and has been exposed to several antibiotics in most cases [23]. Therefore, the patient’s intestinal microbiota has been injured, displays a reduced α-diversity, and is going to receive a conditioning regimen with or without radiotherapy. This means that on a weakened flora susceptible to damage, further depletion will occur. Although we know that chemotherapy impairs intestinal microbiota, the mechanisms by which this occurs are still not completely clear [23,24].

Data about changes in the microbiota composition after HSCT are sometimes conflicting. Holler et al. compared pre- and post-HSCT fecal microbial composition finding an increase in *Enterococci* [20,25] and a reduction in *Firmicutes* and other *phyla* for all patients [20,26]. Recently, Kouidhi et al. reported a comparison between HSCT patients and healthy controls. At the *phylum* level, *Actinobacteria* were more represented in the control group compared with *Proteobacteria* and *Verrucomicrobia* in the HSCT group. At the *genus* level, patients in the HSCT group showed a lower abundance of *Faecalibacterium, Alistipes*, and *Prevotella 9* and a higher abundance of *Bacteroides*, *Escherichia*/*Shigella*, *Klebsiella*, and *Akkermansia* [27]. It is well established that a conditioning regimen involving radioactive sources can lead to dysbiosis of the microbiota, and that radiation-induced enteritis is exacerbated by this dysbiosis. This understanding is derived not only from direct observations of microbiota damage in total body irradiation (TBI) regimens but also from extensive experience in oncology and the widespread use of radiotherapy [28].

In radiation enteritis, as in chemotherapy regimens, an increased abundance of bacteria belonging to the *Actinobacteria* and *Proteobacteria phyla* is reported. These bacteria are conditional pathogens such as *Escherichia coli*. Conversely, microorganisms from the *Firmicutes* and *Bacteroides phyla* are reduced by radiations [29]. In a recent paper, Gu and colleagues reported microbiota changes during myeloablative transplantation, including rabbit thymoglobulin administration. Microbiota diversity began to decline from the start of the conditioning regimen and continued to diminish over the course of HSCT until day 12 after HSCT, where diversity reached the lowest value. After that, diversity gradually increased over time. Intestinal domination varied from beneficial *genus Bacteroides* before conditioning to pathogenic *genera* such as *Enterococcus, Klebsiella*, and *Escherichia* during the engraftment phase [16].

The reduction in microbiota diversity described early after HSCT is a byproduct of conditioning regimens and restoration of pre-HSCT levels is possible over time in the absence of GvHD or other modifying events [27,30]. A reduced microbial diversity after HSCT was associated with high GvHD lethality [14], while the persistence of high microbiota diversity after HSCT was associated with a lower risk of death and TRM, without an increased relapse rate [4]. Results are resumed in Table 2.

### 3.3. Diet Affects Microbiota

The role of diet and nutrition in HSCT patients is often underappreciated, but growing evidence suggests a link between the diet, the microbiota, and clinical outcomes. Although we do not yet know exactly how this element can be used to deliberately select a tolerogenic microbiota, some evidence suggests its potential role in the HSCT setting. It has been demonstrated that enteral nutrition, compared to parenteral nutrition, is protective against the development of GvHD and reduces TRM while increasing OS [31,32].

In this regard, it is even more intriguing evidence provided by Khuat et al., who demonstrated in preclinical models and clinical trials that obesity has a negative effect on HSCT outcomes in both mice and humans. Obesity is specifically associated with an increased risk of aGVHD with GI involvement. This effect appeared restricted to the gut and relies on increased production of pro-inflammatory cytokines by donor CD4+ T-cells. In the murine model, the pre-transplant diet and consequently the selected microbiota can influence TRM, often related to GI aGvHD. Indeed, it has been observed that mice with diet-induced obesity (DIO) exhibited increased gut permeability and translocation of endotoxins across the gut barrier, along with reduced diversity in their gut microbiota. After HSCT, these changes in DIO mice promoted aGvHD and led to a more severe clinical presentation [33].

## 4. Restoration of Microbiota

### 4.1. Prebiotics and Probiotics

The idea that the use of prebiotics could positively influence the microbiota of HSCT recipients was mostly derived from studies on non-hematological patients or retrospective studies [34].

Yoshifuji et al. conducted a prospective study analyzing the clinical effects and changes in the microbiota of patients submitted to HSCT. The study included 30 patients who received a formula called GFO (glutamine, fiber, and oligosaccharide) containing 3 g of glutamine, 5 g of polydextrose, and 1.45 g of lactosucrose. Resistant-starch (RS)-rich dishes, containing 8 g of RS, were provided to patients for lunch and dinner, and one pack of GFO was provided at breakfast from conditioning to day 28. The control group consisted of 142 patients whose data were collected from April 2013 to February 2015. Oral mucositis severity was assessed according to the Eilers Oral Assessment Guide [35]. Although all patients developed mucositis, severity and duration favored the group that consumed prebiotics. Similar results were observed for the development of diarrhea, as the percentage of patients who developed severe diarrhea (grade 3) was comparable between the groups, although in the prebiotic group, 17% of patients did not develop diarrhea compared with 7% in the control group. It is worth noting that there was a 10% lower cumulative incidence of GI-aGvHD [36]. Although further analysis is needed, it appears clear that diet, through its impact on the microbiota, can influence the outcome of an HSCT.

In fact, administering probiotics along with prebiotics as a preventive measure before and during conditioning regimens in patients undergoing HSCT may potentially reduce the occurrence and severity of GVHD by promoting the generation of Tregs, thereby enhancing transplant outcomes [37].

The use of probiotics to promote intestinal well-being is a common practice, but the extent to which this applies to HSCT patients is subject to debate. The use of probiotics in GvHD was always limited by concerns over the development of bloodstream infections (BSI), but a potential beneficial role has already been proposed [38].

Among the most used probiotics are *Lactobacillus* spp. and *Bifidobacterium* spp. A small study by Gorshein et al. demonstrated the safety of probiotics in a small population of 31 patients randomized to receive (n = 20) or not (n = 11) *Lactobacillus rhamnosus*. There was no evidence that the administration of probiotics might change the microbiota of the recipients nor that it could play a role as a therapeutic option in GvHD [39]. The efficacy of probiotics in preventing GvHD is more debated with more assuring data on its safety. It is worth mentioning the work of Yazdandoust and colleagues, in which 40 enrolled patients were divided into two groups: 20 patients received daily symbiotic capsules (*Lactobacillus rhamnosus*, *Lactobacillus casei*, *Lactobacillus bulgaricus*, *Lactobacillus acidophilus*, *Bifidobacterium breve*, *Bifidobacterium longum*, and *Streptococcus thermophilus*), and the remaining 20 who did not receive probiotics or prebiotics [40]. None of the patients in the treated group experienced BSI. Until day +100, two patients (10%) in the intervention group developed aGVHD compared with eight patients in the control group (40%). This study suggested that symbiotic intake might reduce the incidence of GvHD by inducing Treg expansion [41].

The literature aligns with the intuition that an increase in α-diversity is associated with a better clinical outcome, although the current level of evidence is weak. It can be inferred that while probiotics are considered safe and have some evidence of efficacy, it is not incorrect to assume that FMT, being the ultimate probiotic, might provide an even greater clinical benefit.

### 4.2. Fecal Microbiota Transplantation

FMT has been applied to restore intestinal microbial variety damaged by prolonged and combined antibiotics therapy during HSCT. Particularly within this scope, the main concerns involve the risk of infections and GvHD. The occurrence of GvHD varies from 15% to 28% among series [42,43,44], while the risk of GvHD ranges from 6% to 18% [42,43,45].

Rashidi et al. conducted a randomized, phase II, placebo-controlled, double-blind trial (NCT03678493) on FMT as oral capsules for patients with AML and patients submitted to HSCT. The primary aim was to evaluate the incidence of any infection within 4 months of receiving the first FMT course. Seventy-four HSCT recipients were divided into an FMT (n = 49) and a Placebo arm (n = 25). The most frequently documented adverse event (AE) in the FMT arm was GvHD (18.4% vs. 0% in the Placebo arm), followed by bloodstream infection (16.3% vs. 12% in the Placebo arm). A total of 70 infections were documented—43 in the FMT arm and 27 in the other. The infection density within 120 days of the first course of treatment was 0.74 per 100 patient/days in the FMT arm and 0.91 in the other. At the same time point, the mean cumulative number of events per patient was 0.89 and 1.09 in the FMT and Placebo arms, respectively, with an infection rate ratio of 0.83. The 180-day cumulative incidence of grade II–IV GI aGvHD was 25.8% in the FMT arm and 4.3% in the Placebo arm (*p* = 0.03), with a hazard ratio of 5.5. All the cases of grade III–IV GI aGvHD (n = 6) and all fatal cases of aGvHD (n = 3) were documented in the FMT arm. The authors then evaluated microbiota changes before and after treatment administration. The pre-study microbiota composition showed a predominance of *Enterococcus* and *Staphylococcus genera* and a depletion of the *Lachnospiraceae* family, and *Blautia*, *Roseburia,* and *Faecalibacterium genera*. Post-FMT samples showed an increase in *Coriobacteriaceae* and *Rikenellaceae* families compared with *Streptococcus*, *Enterococcus*, *Veionella*, and *Dialister* predominance in the post-placebo samples. A similar recovery in *Blautia* levels was observed in both arms. The α-diversity significantly increased after FMT administration compared with the placebo arm. Therefore, FMT was able to reduce *Enterococcus* and oral bacteria like *Dialister*, and to enhance *Collinsella* and *Blautia* recovery, even if the latter recovered spontaneously in the placebo arm too. Avoiding the administration during neutropenia, FMT did not favor BSI occurrence. Regarding the higher incidence of GvHD in the FMT arm, the author attributed this to the different GvHD prophylaxis schedules among groups [42]. 

The group of Taur designed a randomized controlled clinical trial (NCT02269150) on autologous FMT in HSCT recipients with the aim of restoring gut microbiota, damaged due to antibiotic administration during HSCT. They preferred autologous FMT to avoid recipient exposure to potentially pathogenic microorganisms from a donor. Post-transplant microbiota showed a progressive reduction in diversity indexes, with the lowest value achieved on day +5 after HSCT, persisting for the next 6 weeks. A slow recovery in microbiota diversity was documented after day +50, but it rarely returned to the pre-transplant value. Taur published results on the first 25 patients treated in the trial, which is still ongoing. Patients with microbiota composition which was poor in the *Bacteroidetes phylum* after HSCT were randomly assigned to receive or not at day +49 autologous FMT (harvested before HSCT). On day +21, all patients had markedly reduced microbiota diversity with a predominance of *Enterococcus*. After interventional treatment, patients who had received autologous FMT (n = 14) obtained microbiota diversity and composition restored to pre-transplant levels, while patients in the control arm (n = 11) did not. Commensal groups like *Lachnospiraceae*, *Ruminococcaceae*, and *Bacteroidetes* were successfully recovered after autologous FMT [43].

Another paper by DeFilipp reported results from a single-arm study on FMT from healthy donors administered as oral capsules within 4 weeks of neutrophil engraftment (NCT02733744). Among thirteen treated patients, only one AE was reported (grade 3 abdominal pain). However, two patients developed grade III–IV GI-aGvHD, one of whom also experienced a BSI sustained by *Klebsiella pneumoniae* with consequent multi-organ failure. Another patient developed *Clostridium difficile* colitis. Moreover, six patients developed moderate–severe chronic GvHD. One-year OS was 85%. The authors also evaluated the urinary concentration of 3-indoxyl-sulfate (3-IS) as a surrogate marker for microbiota disruption. The 3-IS level significantly decreased from the pre-HSCT to post-HSCT period but then increased again after FMT. *Clostridiales* abundance decreased from pre-HSCT to post-HSCT samples, but it was restored via FMT. After FMT, the recipient microbiota composition was similar to that of the donor, as demonstrated by the operational taxonomic unit (OTU) origin analysis [46].

Dougè et al. published their multicentre, randomized, phase II clinical trial protocol on FMT administration in patients submitted to myeloablative HSCT (NCT04935684). The primary aim is 1-year GvHD-free relapse-free survival (GRFS) rate. Secondary aims are outcome measures of the impact of FMT on HSCT-related morbidity and mortality. Within 4 weeks of neutrophil engraftment, patients will be randomly assigned to receive (n = 60) or not receive (n = 60) healthy, donor-derived FMT as an enema, to be administered via rectal cannula [47]. Table 3 summarizes studies on FMT applied for microbiota restoration after HSCT.

## 5. Microbiota Changes Associated with GvHD

The GI microbiota has been reported to be implicated in the development of aGvHD. Specifically, the damaged GI epithelial barrier in HSCT patients allows the translocation of microorganisms or pathogen-associated molecular patterns (PAMPs) [48]. These molecules can activate APCs, leading to the activation and proliferation of alloreactive donor T cells that primed aGvHD [49]. Damage to the microbiota due to chemotherapy or broad-spectrum antibiotics results in excessive exposure of the organism to PAMPs and pathogens, triggering an amplified immune response. Recent evidence suggests that these mechanisms may serve as prodromal factors for GvHD [22].

Disruption of the intestinal milieu due to the depletion of specific bacterial subsets by antibiotics such as carbapenems may disturb the equilibrium of monosaccharide concentrations, consequently influencing the behavior of commensal mucolytic bacteria. Consequently, this perturbation may contribute to an escalated manifestation of GvHD [18]. An Italian group reported that stool samples collected at day +10 after HSCT were highly informative regarding the risk of developing GvHD. An overexpression of *Enterococcaceae* and a reduction in *Lachnospiraceae* resulted to be predictive for grade II–IV aGvHD and for GI involvement [50,51], while a predominance of *Staphylococcaceae* predicted liver and GI involvement and steroid-resistance of aGvHD [51,52]. Patients who developed GI GvHD showed an almost complete loss of commensal *Firmicutes* with overgrowth of *Enterococcus*, but once resolved, microbiota returned to be similar to that pre-transplant [20]. The correlation between *Enterococcus faecalis* and GvHD was tested in mice. Pretransplant colonization of mice with *Enterococcus faecalis* showed that it also remained detectable in the first week after transplantation. Mice harboring *Enterococcus faecalis* experienced more severe GvHD and showed high levels of interferon-ɣ, increased number of donor T cells, proliferating T CD4+, and increased number of T helper-17 (Th17) in the colonic lamina propria [53]. Similarly, the administration of *Enterococcus faecalis* to mice after HSCT aggravated GvHD. Moreover, enterococcal expansion after HSCT was associated with *Clostridium* loss. This is important because *Clostridium* is known to produce butyrate, to increase the number of immune-regulatory macrophages in the GI tract with the subsequent increased number of Tregs [54], and to contribute to better survival and low incidence of GvHD when strongly represented in the gut [25,55]. For Doki et al., an abundance of *Firmicutes* in pre-HSCT samples was predictive for aGvHD [56]. Analyzing stool samples obtained at the onset of aGvHD, our group showed a lower abundance of *Bacteroidetes* and a higher abundance of *Firmicutes* and *Proteobacteria*, although patients with GI aGvHD retained a higher presence of *Bacteroidetes* compared with patients with liver or skin involvement [26]. A comparative study between patients with GvHD, patients without GvHD, and healthy controls showed a high abundance of *Escherichia/Shigella* and *Bacteroides* in the first group, *Klebsiella*, *Akkermansia*, *Lachnospiraceae*, and *Veillonella* in the second group and *Prevotella 9*, *Alistipes*, and *Faecalibacterium* in the last group, respectively [27].

Han et al. created and validate a gut microbiota score (GMS) able to predict aGvHD after myeloablative HSCT. Pre- and post-HSCT stool and blood samples were collected from a discovery cohort of 102 patients and a validation cohort of 48 patients. Stool analysis revealed a less complex taxonomic composition among patients who developed aGvHD. The relative abundances of *Lachnospiraceae* and *Peptostreptococcaceae* were negatively correlated with aGVHD occurrence, whereas the relative abundance of *Enterobacteriaceae* was positively correlated with aGVHD development. GMS was associated with the inverse Simpson index, which was higher in the low-GMS group. The inverse Simpson index is an ecological estimate of diversity calculated to represent the reciprocal of the expected probability of randomly selected bacterial sequences belonging to the same OTU [15]. The cumulative incidence of grade II–IV aGvHD was lower in the low-GMS group compared to the high-GMS group (27.1% vs. 84.8%). Similarly, the incidence of grade III–IV aGvHD was 4.3% and 30.3% in the low- and high-GMS groups, respectively. Multivariate analysis revealed high GMS, inverse Simpson index, intensified conditioning and β-lactam antibiotics were independent risk factors for grade II–IV aGvHD. The authors also investigated the association between microbiota and Treg and Th17. After HSCT, the Treg count was higher in the low-GMS group compared with the high-GMS group, as for the ratio Treg/Th17, while the Th17 count was lower in the low-GMS group than in the high-GMS group [57]. Liu et al. hypothesized that GI aGvHD might be promoted by differences in microbiota composition between recipient and donor. In fact, intestinal bacteria and their metabolites influence immune response through T lymphocyte activation and or T cell differentiation toward Treg [58,59,60]. Therefore, when exposed to the recipient microbiota, donor immune cells might trigger the GvHD process. Starting from that point, the authors analyzed the microbiota composition in a cohort of 57 patients and 22 paired HLA-matched sibling donors. Stool samples were collected within the 7 days before starting conditioning chemotherapy in patients or granulocyte-colony-stimulating factor (G-CSF) administration in donors. Pre-HSCT recipient microbiota showed low microbial diversity with an abundance of facultative anaerobic bacteria such as *Enterobacteriaceae*, *Lactobacillaceae*, *Enterococcaceae*, and *Streptococcaceae*, while in donor microbiota, obligate anaerobes such as *Bacteroidaceae*, *Lachnospiraceae*, and *Ruminococcaceae* were more represented. Interestingly, donor but not recipient microbiota composition were reported to affect GI aGvHD [61]. Thus, the high diversity of donor microbiota would contribute to greater tolerance by donor alloreactive T lymphocytes resulting in reduced GvHD. Moreover, the abundance of *Parabacteroides distasonis* and *Barnesiellaceae* in recipient microbiota reduced the risk of GvHD, probably due to their anti-inflammatory role [62,63,64]. We are aware that certain bacterial species, such as Eubacterium *Limosum* and *Blautia*, have a protective role in GvHD mechanisms. These species exhibit an anti-inflammatory phenotype by producing SCFAs like butyrate [65,66]. *Blautia* was reported as closely associated with GvHD outcomes. Its abundance resulted associated to reduced GvHD-related mortality and reduced incidence of clinically relevant GvHD needing steroid treatment [14,45,50]. Among patients with GvHD, the *Blautia genus* showed the highest number of interactions with other bacteria, despite its relatively low abundance [27]. *Blautia* abundance appeared to be significantly compromised by antibiotics with anaerobic coverage and in patients who received prolonged total parenteral nutrition [14]. Bacterial metabolites have also been studied in the HSCT setting. After transplant, a reduced amount of SFCAs, acetate, propionate, and butyrate were reported [50,67] but they progressively recovered over time to pre-HSCT levels [67]. The altered pathway of the fecal metabolome in HSCT patients showed a predominance of lipids, particularly fatty acids, along with succinic acid and fumaric acid [27].

### Fecal Microbiota Transplantation for GvHD

Several studies were conducted on the use of FMT in the HSCT setting [68,69,70].

Fecal microbiota is usually obtained by healthy donors who have passed clinical and microbiological screening. In a minority of cases, relatives or consorts were used for microbiota donation. The way to administer fecal microbiota may vary from stool suspension infused as a rectal enema or via nasoduodenal or nasojejunal tube, or it can be administered orally as capsules. The primary indication was steroid-resistant (SR) or steroid-dependent (SD) GI aGvHD. The most frequently reported side effects attributable to FMT were GI symptoms such as abdominal pain, bloating, and diarrhea. Beyond the response rate, the main concern was about the risk of infectious complications. In fact, usually, FMT was administered once neutrophil engraftment is obtained for reducing the risk of bacterial infections during the neutropenic phase, and in the absence of clinically relevant GI symptoms or toxicity to limit luminal bacterial spread through the damaged intestinal barrier. Finally, to avoid an impairment in the richness and diversity of the infused microbiota, antibiotics should be discontinued at least 48 h before FMT and should not be administered for 48 h after FMT. SR/SD GI aGvHD response rates to FMT varied from 28% to 75% among series [44,45,71,72]. Single case reports were published [73,74,75] but we have focused on case series.

A small study by Kakihana et al. reported the administration of FMT via nasoduodenal tube from a wife or a relative in four patients with SR/SD GI aGvHD receiving methylprednisolone. A complete response was obtained in three patients allowing consistent reduction of steroid therapy, while a partial response was obtained in the fourth patient. In patients obtaining a complete response, restored microbiota showed a predominance of *Bacteroides*, *Lactobacillus*, *Bifidobacterium*, and *Faecalibcterium* [71].

Another small cohort of three patients with SR GI aGvHD was treated for compassionate use with repeated instillation of FMT from healthy donors via a colonoscope. Two patients obtained a complete remission while the third obtained only a transient improvement of the diarrhea. All the patients died after FMT, but death causes were not considered attributable to the FMT itself [76].

Goeser reported two German center case series (n = 11) of FMT administration via nasojejunal tube or as oral capsules as rescue therapy for GI SR-GvHD. Six patients were also receiving ruxolitinib at the time of FMT. Mild GI symptoms due to FMT were recorded as AE in 5 cases, while no severe AE was seen. Fourteen days after FMT, stool frequency and volume significantly diminished, as for C reactive protein levels which were found to be correlated with stool volume and frequency. Pre- and post-FMT microbiota analysis revealed a reduced ɑ-diversity before FMT, but it increased after FMT, although it never reached donor values. The β-diversity analysis showed that pre-FMT, post-FMT, and donor microbiota samples clustered separately with a swift of post-FMT samples composition toward donor pathways. The microbiota composition analysis showed that FMT led to an increase of *Ruminococcaceae*, *Bacteroidaceae*, *Lachnospiraceae*, *Streptococcaceae*, and *Lactobacillaceae*, while it reduced *Akkermansiaceae*, *Enterococcaceae*, *Veionellaceae*, *Peptostreptococcaceae*, and *Clostridiaceae* [77].

In a pilot study by Shouval et al. (NCT03214289), seven patients with SR/SD GI aGvHD received one to three courses of FMT from unrelated donors as oral capsules. Two patients developed BSI from a pathogen not detected in the transplanted microbiota. A complete response of GvHD was documented in 2/7 patients. Four deaths were documented, three of which were due to progressive GvHD and the other to infectious complications in a patient who had obtained a GvHD remission after FMT. Four patients showed a predominance of *Escherichia coli* before FMT and its reduction after treatment [44].

Zhao and colleagues published the results of their non-randomized, open-label, phase I/II trial (NCT03148743) on 41 patients with SR GI aGvHD. A total of 23 patients who received FMT were compared with 18 patients as a control group. FMT collected from four healthy donors was administered via nasojejunal tube. On day 14 from SR GI aGvHD, 52.2% of patients in the FMT group obtained a clinical remission and none in the control group did, while an overall response was registered in 82.6% and 39% in the two groups, respectively. On day 21, 56.6% of patients in the FMT group and 16% in the control group had obtained clinical remission without differences in clinical response. Three patients in the FMT group and two patients in the control group died. Two patients from each group showed GI GvHD relapse. Overall, no difference was seen in terms of event-free survival between groups, while OS was better in the FMT group (HR 4.2). Severe AE in the FMT group were thrombocytopenia and cardiac events, while moderate AEs were reported in six other patients (gastrointestinal symptom, fever, rash). The low microbiota diversity in recipients including a higher abundance of *Proteobacteria* and a lower presence of *Firmicutes* was restored after FMT. *Bacteroidetes* were more represented in stool samples from patients with SR GI aGvHD. After FMT, *Firmicutes* abundance increased while *Proteobacteria* decreased [72,78]. Subsequently, Liu et al. [79] described a subset cohort of the previous trial including 21 patients with grade III–IV SR GI aGvHD treated with FMT and ruxolitinib. The overall response rate was 71.4% after a median of 10 days, with 10 complete responses and 5 partial responses, and a median time to steroid tapering to half dose of 14 days. Durable responses were documented in 80% of responders. A GvHD relapse occurred in a third of cases. The most frequent AEs were viral reactivations (62%), bacterial infections (29%), and severe cytopenias (81%). A reduction in inflammatory cytokines such as interleukin (IL)-2 and IL-17A and activated T cells along with an increase in Tregs was observed in responders. Moreover, an increase in *Lactobacillus* with a reduction of *Escherichia* was also reported in responders [80]. 

In another study, Van Lier and colleagues enrolled 17 patients submitted to HSCT receiving FMT via nasoduodenal infusion for SR or SD grade II–IV GI aGvHD. The main reported AEs were gastrointestinal symptoms that resolved spontaneously within hours. Five patients developed an infection within the first month after FMT: otitis media (n = 1), cystitis (n = 2), pneumonia (n = 1), and sepsis (n = 1). A complete response of GvHD was obtained in 50% SR GI aGvHD and 78% SD GI aGvHD. Response to FMT was associated with lower GvHD grade at treatment. Among patients who obtained a complete response, 6 patients maintained the response after immunosuppression suspension, while 4 patients showed a secondary failure. Four of the five non-responders patients died because of GvHD complications. Among responders, pre-FMT *Blautia* abundance was comparable to that in the donors, and it increased further after successful FMT treatment. Similarly, *Clostridiales* and butyrate producers gradually increased in responders achieving levels of the donor, while these bacteria remained stable or declined in non-responders [45]. Recently, Malard et al. published the results of the use of pooled allogeneic microbiotherapeutic MaaT013 in patients with SR GI aGvHD. MaaT013 is a preparation with high OTU richness and high microbial diversity, administered via enema. They included 24 patients from the prospective, single-arm phase 2 HERACLES study (NCT03359980) and 52 patients from the expanded access program (EAP). All patients received at least two doses of MaaT013. Thirteen patients in the HERACLES cohort and thirty-five from the EAP obtained at least a partial response. In the HERACLES group, only five serious AE were reported, which were infectious complications due to a pathogen not detected in the MaaT013. Microbiota composition analysis revealed an increased richness and α-diversity after treatment at any time point [79].

Active clinical trials on FMT treatment for SR/SD GvHD are reported in Table 4.

## 6. Fecal Microbiota Transplantation for Bacterial Infection Treatment or Prophylaxis

HSCT patients exhibit greater exposure to antibiotics compared to autologous stem cell transplanted patients, and this heightened exposure is associated with a diminished pre-transplant α-diversity [81,82]. This assumption is directly linked to the most recent finding that the incidence of *Clostridium difficile* infection (CDI) after HSCT is 6% to 20%, consistently higher than that seen in the autologous stem cell transplant setting (5–6%) [83,84,85]. The microbiota dysbiosis is a prerequisite to CDI and patients with primary CDI often have a minor degree of dysbiosis when compared to patients with recurrent CDI [86]. Notably, the European guidelines for CDI do not specifically mention transplant patients as a distinct category, although the epidemiology and comorbidities of these patients make us understand how relevant the issue is. As of the latest update in 2021, the treatment for CDI remains preferentially with fidaxomicin (first-line, strong recommendation, moderate level of evidence) and vancomycin (second-line) [87,88]. For the first recurrence, the combination of bezlotuximab is recommended. It is essential to mention that FMT is indicated only for the second recurrence [89]. We have data indicating that FMT appears to be both a safe and effective treatment of CDI in recipients of HSCT [90]. The Moss team worked in this direction by studying the microbiome of eight patients with recurrent CDI treated with FMT. The investigators reaffirm the advantage of FMT in a careful risk–benefit assessment of this patient population, along with its safety [91]. Furthermore, in the one-year follow-up, they observed a differentiation of the microbiome between the donor and the recipient, meaning that the microbiota engraftment is heavily impacted by other factors such as diet, host genetics, host immune surveillance, and subsequent antibiotic exposure. These factors may supersede FMT composition in determining long-term FMT durability [92]. Currently, there is only one study that investigates this issue [43], as reported in Table 5.

Pre-HSCT microbial composition is predictive for infectious complications after HSCT [94], with particularly poor outcomes reported in patients colonized with multidrug-resistant (MDR) pathogens [95]. Although knowledge of pre-HSCT colonization can guide antibiotic choice during febrile neutropenia, the mortality rate related to BSI induced by MDR bacteria remains high [96,97]. Several authors have evaluated the possible role of FMT in reducing MDR-related BSI in the HSCT setting.

Ghani et al. conducted a study on FMT to prevent invasive infections induced by colonizing MDR bacteria. Twenty patients from five London centers received FMT via nasogastric tube and compared with a similar cohort that did not receive FMT. A significant reduction in BSI was observed in the treated cohort, as well as being documented a reduced inpatients stay post-FMT and a reduced need for carbapenems use. FMT secured decolonization in 41% of patients. Mild and self-limiting GI symptoms were observed after FMT, without serious events occurrence. Among the treated cohort, 8 patients underwent HSCT. In the post-HSCT period, a reduced hospital stay and a minor number of carbapenems days of treatment were observed as compared with the 6 months preceding HSCT. Only one patient developed a BSI induced by an MDR pathogen different from the original colonizing [98].

Battipaglia and colleagues evaluated the efficacy of FMT in eradicating MDR bacteria colonization in hematological patients before or after HSCT with the purpose to reduce infectious complications. Vancomycin-resistant *Enterococci*, carbapenemase-producing *Enterobacteriaceae*, or carbapenemase-producing *Pseudomonas aeruginosa* in at least three consecutive microbiological cultures were considered suitable for FMT [99]. Among 10 treated patients, 6 received FMT at a median time of 163 days after HSCT. The FMT donor was a relative in half of the cases and unrelated in the others, and the way of FMT administration was enema in 4 cases and nasogastric tube in the others. Three patients required a second course of FMT. Globally, three patients obtained major decolonization and two patients died because of underlying SR GI GvHD in one case and disease progression in the other. No major AEs were registered but GI symptoms related to FMT were observed in three patients [100].

Following a previous study in which colonized patients underwent HSCT reported high infectious-related mortality and high non-relapse mortality with overall worse survival [101]. Bilinski et al. conducted a prospective study. Twenty patients with intestinal MDR bacteria colonization, including patients who received HSCT, were treated with FMT administration via a nasoduodenal tube. Predominant MDR colonizing bacteria were *Klebsiella pneumoniae*, *Enterococcus*, *Escherichia coli*, *Enterobacter*, and *Pseudomonas aeruginosa*. Complete decolonization was observed in 60% of patients one month after FMT, with a predominance of patients who had not received antibiotics after FMT administration. GI symptoms were frequently documented as associated with FMT, particularly diarrhea. In the follow-up period, there were two documented cases of sepsis, two cases of recolonization with the same bacteria, and a case of colonization with a new bacteria strain [102].

Innes et al. conducted a retrospective study on FMT in patients with MDR bacterial colonization before receiving HSCT. Nineteen patients with MDR bacteria colonization candidates to receive HSCT were enrolled. Eight patients received FMT before HSCT while eleven patients did not. The probability of survival at 12 months was 70% in the FMT group and 36% in the non-FMT group. A clinical infection was the cause of death in one patient in the FMT group and in five patients of the non-FMT group. The number of days of fever normalized for the number of admission days was lower in the FMT group (0.11 vs. 0.29). Decolonization was obtained in 25% of the patients in the FMT group compared with 11% spontaneous decolonization in the no-FMT group [94]. Table 5 resumed active studies on FMT for CDI or MDR colonization.

## 7. Conclusions

Over the past decade, FMT has been extensively studied in various clinical settings. Among them, the setting of HSCT has attracted much interest because of the different indications for the application of FMT: reconstitution of the microbiota damaged by chemotherapies and antibiotics, treatment of GvHD, prevention of severe infections sustained by multidrug-resistant bacteria, and prevention and treatment of *Clostridium difficile*. Undoubtedly, our review has shown heterogeneity in study design, route of administration and dosage schedules used, and enrolled populations. However, FMT has demonstrated variable efficacy in the different fields of application, but more importantly, it has greatly allayed the concerns regarding the infectious risk associated with the procedure. Inevitably, the role of FMT in the development of GvHD when FMT is used for other indications remains to be clarified. Despite the currently encouraging results in terms of efficacy and safety, further studies in larger patient cohorts are needed to confirm the reported results and to clarify the real impact of FMT on the development of GvHD, both when FMT is used as salvage therapy for GVHD SR, but especially when used for preventing infections.

## Figures and Tables

**Figure 1 microorganisms-11-02182-f001:**
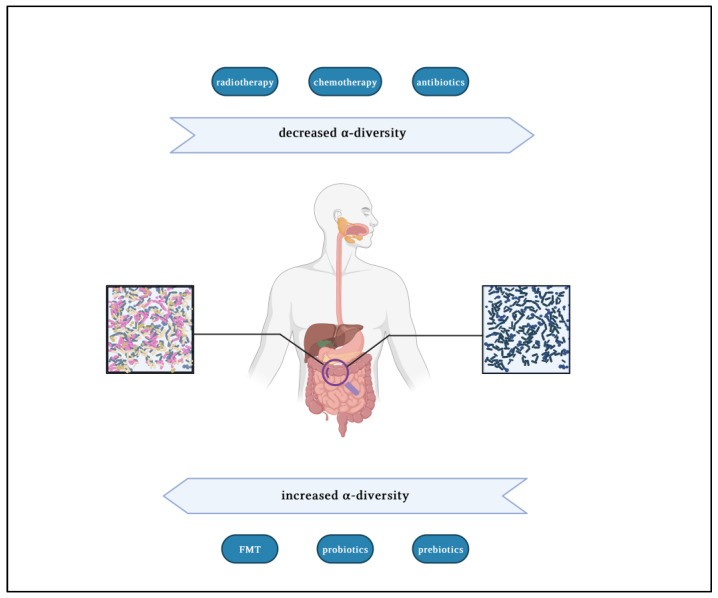
The major factors involved in reducing or amplifying the α-diversity in patients undergoing allogeneic stem cell transplantation. Created with “BioRender.com” accessed on 13 of August 2023 (L.D.M. obtained Publication and Licensing rights by BioRender with agreement number YP25Q4SXGS).

**Table 1 microorganisms-11-02182-t001:** Microbiota changes due to conditioning regimen.

References	Population	Antibiotic	Microbiota Changes	Effect
Hayase et al. [18]	Mice	Cefepime and Levofloxacin vs. meropenem	↑ *Clostridiales* ↓ *Bacteroides thetaiotaomicron* *↑ Enterococcus*	↓ severe GvHD
Shono et al. [19]	T-cell-replete HSCT human and mice	Piperacillin-tazobactam Imipenem-cilastatin Azteronam and Cefepime	↓ *Actinobacteria* ↓ *Clostridiales ↑ Erisipelotrichia* and *Enterococcus* ↔ *Clostridiales*	↑ grade II–IV aGvHD ↑ GvHD-related mortality ↓ GvHD-related mortality
Holler et al. [20]	Human HSCT	Ciprofloxacin and broad-spectrum antibiotics	↑ *Enterococci* ↓ classical commensal bacteria	↑ GI-GvHD
Weber et al. [21]	Human HSCT	Ciprofloxacin and metronidazole vs. rifaximin	↔ *Enterobacteriaceae*	↓ GI-GvHD ↓ TRM ↑ OS

GvHD: graft-versus-host disease; HSCT: hematopoietic stem cell transplantation; GI: gastrointestinal; TRM: transplant-related mortality; OS: overall survival; ↑:increased abundance; ↓: reduced abundance; ↔: unmodified abundance.

**Table 2 microorganisms-11-02182-t002:** Microbiota changes due to conditioning regimen.

References	Population	Microbiota Changes	Effect
Stein-Thoeringer et al. [25]	Human and mice HSCT	*↑ Enterococcus genus and Enterococcus faecium species*	↓ OS *↑* GvHD-related mortality
Chiusolo et al. [26]	Human HSCT	*↑ Bacteroidetes* *↓ Firmicutes*	- -
Kouidhi [27]	HSCT vs. Healthy subjects	*↑ Proteobacteria* and *Verrucomicrobia phyla* *↓ Faecalibacterium, Alistipes*, and *Prevotella 9 genera* *↑Bacteroides, Escherichia/Shigella,* *↑ Klebsiella, Veionella* and *Akkermansia genera* *↑ Actinobacteria phylum* *↑ Faecalibacterium, Alistipes*, and *Prevotella 9 genera*	- - *↑ GvHD* *↓ GvHD*
Jian et al. [28]	Radiotherapy	*↓ Lactobacillus and Bifidobacterium* *↑ Staphylococcus and Escherichia coli*	↑ radiation enteritis ↓ intestinal barrier function ↑ inflammation
Gu et al. [16]	Myeloablative conditioning plus ATG	*↓ Bacteroides genus and* *↑ Enterococcus, Klebsiella, and Escherichia genera*	↓ OS for low microbial diversity

HSCT: hematopoietic stem cell transplantation; GvHD: graft-versus-host disease; ATG: anty-thymocyte globulins; OS: overall survival; ↑:increased abundance; ↓: reduced abundance.

**Table 3 microorganisms-11-02182-t003:** Active or completed studies about fecal microbiota transplantation for restoring microbiota after HSCT.

Publication	Registration Number	Indications	Phase	Number of Patients	Age, Years	Intervention	Adverse Events	Status
DeFilipp [46]	NCT02733744	HSCT	Early phase 1	13	18–65	FMT as oral capsules	1 AE 2 GvHD 1 Sepsis 1 CDI	Completed
Rashidi [42]	NCT03678493	AML and HSCT	Phase 2	45 AML 74 HSCT	18 and older	FMT as oral capsules vs. placebo	GvHD 18.4% (FMT arm) vs. 0% (placebo arm) BSI 16.3% (FMT arm) and 12% (placebo arm)	Active not recruiting
Dougè [47]	NCT04935684	Myeloablative HSCT	Phase 2	150	18 and older	FMT vs. placebo as enema via rectal cannula	Not applicable	Recruiting
unpublished	NCT03720392	Myeloablative or intermediate intensity HSCT	Phase 2	8	18–80	FMT as oral capsules vs. placebo	1 sepsis (FMT arm)	Completed
Taur [43]	NCT02269150	HSCT	Phase 2	25	18 and older	FMT via enema vs. placebo	Not available	Active not recruiting

Legend: HSCT: allogeneic hematopoietic stem cell transplantation; FMT: fecal microbiota transplantation; AE: adverse event; GvHD: graft-versus-host disease; CDI: clostridium difficile infection; AML: acute myeloid leukemia.

**Table 4 microorganisms-11-02182-t004:** Active studies about fecal microbiota transplantation for GvHD.

Registration Number	Indications	Phase	Number of Patients	Age, Years	Intervention	Administration Way	Allocation	Status
NCT04711967	SD/SR gut aGvHD	Not applicable	20	18–60	FMT vs. no treatment	unknown	Randomized	Recruiting
NCT04935684	HSCT	2	150	18 and older	FMT vs. no treatment	250 mL of enema	Randomized	Recruiting
NCT04139577	Grade II–IV aGvHD and high risk naïve aGvHD	1	10	18 and older	FMT	40 oral capsules	Single-arm	Active not recruiting
NCT04269850	Grade III–IV GI aGvHD	2	20	5–70	FMT	Ruxolitinib 10 mg twice a day, MP 0.5 mg/Kg, 2 FMT capsules/Kg	Single-arm	Recruiting
NCT03812705	SD/SR GI aGvHD	2	30	14–60	FMT	200–300 mL fecal microbiota	Single-arm	Recruiting
NCT03819803	GI aGvHD	3	15	18 and older	FMT	200 mL of enema	Single-arm	Recruiting
NCT03148743 [72,78,79]	SR GI aGvHD	Not applicable	50	10–60	FMT	200 mL of enema	Single-arm	Recruiting
NCT04769895	SR/Ruxolitinib refractory GI GvHD	3	75	18 and older	FMT	Enema	Single-arm	Recruiting
NCT04745221	HSCT	Not applicable	100	18–60	FMT	Oral capsules	Single-arm	Recruiting

Legend: SD: steroid-dependent; SR: steroid-refractory; aGvHD: acute graft-versus-host disease; FMT: fecal microbiota transplantation; HSCT: allogeneic hematopoietic stem cell transplantation; GI: gastrointestinal; MP: methylprednisolone.

**Table 5 microorganisms-11-02182-t005:** Active studies about fecal microbiota transplantation for CDI and MDR infections.

Registration Number	Indications	Phase	Number of Patients	Age, Years	Intervention	Administration Way	Allocation	Status
NCT02269150 [43]	HSCT	2	59	18 and older	FMT	FMT via enema vs. placebo	Single-arm	Active not recruiting
NCT02461199	Blood diseases	Not applicable	50	18 and older	FMT	100 mL of enema	Single-arm	Recruiting
NCT04431934	MDR carriers	Not applicable	437	18 and older	FMT vs. Probiotics	14–17 FMT oral capsules vs. 2 sachets of probiota	Randomized	Recruiting
NCT03167398 [93]	CRE carriers	2	15	18 and older	FMT	30 FMT oral capsules	Single-arm	Completed

Legend: HSCT: allogeneic hematopoietic stem cell transplantation; FMT: fecal microbiota transplantation; MDR: multidrug-resistant; CRE: carbapenemase-resistant Enterococcaceae.

## Data Availability

The list of articles we have read and analyzed can be made available to readers by email request to the corresponding author.

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
