# Peer review of "The Role of Fecal Microbiota Transplantation in the Allogeneic Stem Cell Transplant Setting"

_microorganisms, 2023, doi:10.3390/microorganisms11092182_

Round 1

Reviewer 1 Report

Abbreviations should be stated at the first mention.

Lines 38-41: The authors should design a table listing most microbial groups and strains prevalent at different ages.

Regarding the main factors affecting gut microbiology, it will be easy for those interested in the subject to use tables that summarize the latest results, for example, "antibiotics." A table showing the type of antibiotics, age, duration of treatment, type of effect, and reference

Minor editing.

Author Response

Comment: Abbreviations should be stated at the first mention.

Reply: Thanks. We added the abbreviation explanations where they were missing

Comment: Lines 38-41: The authors should design a table listing most microbial groups and strains prevalent at different ages.

Reply: this review is focused on fecal microbiota transplantation in the allogeneic stem cell transplantation setting. Microbiota differences at different age is not pertinent with the contents.

Comment: Regarding the main factors affecting gut microbiology, it will be easy for those interested in the subject to use tables that summarize the latest results, for example, "antibiotics." A table showing the type of antibiotics, age, duration of treatment, type of effect, and reference

Reply: we added tables resuming the main findings for each section.

Reviewer 2 Report

Overall, it is a fine manuscript focusing on The role of fecal microbiota transplantation in allogeneic stem cell transplantation, which is a quite specific field that lacks comprehensive reviews. So, in terms of the topic, it is very interesting and may attract more attention to allogeneic hematopoietic stem cell transplantation. The authors well-summarized the microbiota changes during allogeneic stem cell transplantation that are associated with the antibiotic application, conditioning, and diet effects, as well as focused on the potential methods for the restoration of microbiota, which may guide future therapeutics. Specifically, as a serious complication, more research is required for GvHD treatment. The authors also well-summarized this part including the up-to-date development and findings. However, this part needs to be reorganized into separate paragraphs.

Figure 1 quality needs to be improved.

Author Response

Comment: Overall, it is a fine manuscript focusing on The role of fecal microbiota transplantation in allogeneic stem cell transplantation, which is a quite specific field that lacks comprehensive reviews. So, in terms of the topic, it is very interesting and may attract more attention to allogeneic hematopoietic stem cell transplantation. The authors well-summarized the microbiota changes during allogeneic stem cell transplantation that are associated with the antibiotic application, conditioning, and diet effects, as well as focused on the potential methods for the restoration of microbiota, which may guide future therapeutics. Specifically, as a serious complication, more research is required for GvHD treatment. The authors also well-summarized this part including the up-to-date development and findings. However, this part needs to be reorganized into separate paragraphs.

Reply: we tried to make this part more readable.

Figure 1 quality needs to be improved.

Reply: we created an improved figure using bioRender.com

Reviewer 3 Report

Dear Authors,

The submitted review addresses the actual and challenging topic for numerous peers from the fundamental and clinical disciplines. It is comprehensive, competent, and well-structured. The careful evaluation yielded the necessity for a minor revision before the acceptance for publication. Please find below the detailed comments and suggestions.

Title

- Please consider the additional efforts to avoid repeating the term transplantation.

Abstract

-It would be relevant to quote PubMed as the data source.

Keywords

- The abbreviations do not seem suitable as the keywords.

Introduction

- Line 62: The abbreviation HSCT does not fully correspond to the explanation.

- Line 63: The term conditional regimen is not precisely explained.

- Line 71: The readership would benefit from a brief explanation of the term α-diversity of fecal microbiota.

- Lines 97–106: To enhance understanding, please consider including the explanation that Clostridias represent beneficially and Bacteroides thetaiotaomicron detrimental bacteria for mucus integrity.

Conditioning affects gut microbiota

- At some points, the text is challenging to follow due to the overlap of the information about the alterations in gut microbiota before and after the whole transplantation process and changes related to the conditional phase (mainly irradiation) only. Please consider restructuring to improve this aspect.

- To complete the perception in this section, including the description of the changes associated with the other conditioning modalities, like chemo- or immunotherapy, would be appreciated.

Fecal microbiota transplantation

- What was the precise meaning of marking the study from Reference 46 as the early phase 1 study?

- After section 3. Restoration of microbiota, please consider including a summary paragraph that would provide data about the eventual guidelines on this aspect, together with the current state of the art regarding fecal microbiota transplantation. The suggestion also refers to sections 4 and 5.

Microbiota changes associated with GvHD

- Line 345: Please include the explanation for the Simpson index.

- It would be appreciated to include the achievements about using pre- and probiotics in GvHD. The suggestion also refers to the section 5.

Fecal microbiota transplantation for GvHD

- Line 401: Citing three publications does not support the information about many case reports.

Technical suggestions

- Please consider the efforts to explain the abbreviation when you introduce it.

- Additional caution is necessary to use the italic font for the names of microorganisms and the Latin terms.

- Please replace the abbreviations bid and tid with the terms familiar to the broad readership.

The language editing by the professional service or the native speaker would additionally improve the overall quality of the Manuscript.

Author Response

Title

Comment: Please consider the additional efforts to avoid repeating the term transplantation.

Reply: we have modified the title.

Abstract

Comment: It would be relevant to quote PubMed as the data source.

Reply: we added in the abstract PubMed data source

Keywords

The abbreviations do not seem suitable as the keywords.

Reply: we have modified keywords as suggested by the reviewer.

Introduction

Comment: Line 62: The abbreviation HSCT does not fully correspond to the explanation.

Reply: we have modified the explanation of abbreviations.

Comment: Line 63: The term conditional regimen is not precisely explained.

Reply: actually "conditioning regimen" is a hematology-specific and universally used term to refer exclusively to chemotherapy that precedes the stem cell infusion in the transplant procedure

Comment: Line 71: The readership would benefit from a brief explanation of the term α-diversity of fecal microbiota.

Reply: we have specified the term α-diversity.

Comment: Lines 97–106: To enhance understanding, please consider including the explanation that Clostridias represent beneficially and Bacteroides thetaiotaomicron detrimental bacteria for mucus integrity.

Reply: we have added the explanation at the end of the period.

Conditioning affects gut microbiota

Comment: At some points, the text is challenging to follow due to the overlap of the information about the alterations in gut microbiota before and after the whole transplantation process and changes related to the conditional phase (mainly irradiation) only. Please consider restructuring to improve this aspect.

Reply: Conditioning regimen is the preparative therapy administered before stem cell infusion during the transplant procedure. Conditioning regimen might include chemotherapy only or a combination of chemotherapy and radiation. Radiation only is not usually considered a standard conditioning regimen. Therefore, it is difficult to divide the effects of radiation compared to the effects of chemotherapy because the available paper in the hematological setting did not distinguish between those.

Comment: To complete the perception in this section, including the description of the changes associated with the other conditioning modalities, like chemo- or immunotherapy, would be appreciated.

Reply: we added another study that reported microbiota changes during transplantation performed with a conditioning regimen that included thymoglobulin as immunotherapy. However, even in this case there were no differences with conditioning regimen that do not include thymoglobulin, and patients enrolled had received a conditioning regimen with chemo only or a combination of chemo and radiotherapy. I know that the transplant setting is very complex and difficult to explain and much more to understand. Differing from solid organ transplantation, patients submitted to HSCT receive a combination of many drugs in the few days preceding the stem cell infusion, a combination of antibiotics in the pre-engraftment period, and a combination of many immunosuppressant drugs from the stem cell infusion over the subsequent one to six months. Therefore, we highlighted only differences more certainly associated with one of the reported risk factors for those papers comparing different groups.

Fecal microbiota transplantation

Comment: What was the precise meaning of marking the study from Reference 46 as the early phase 1 study?

Reply: we simply reported studies’ characteristics registered on the official clinicaltrial website.

Comment: After section 3. Restoration of microbiota, please consider including a summary paragraph that would provide data about the eventual guidelines on this aspect, together with the current state of the art regarding fecal microbiota transplantation. The suggestion also refers to sections 4 and 5.

Reply: unlike the gastroenterological setting, to date, there are no guidelines or suggestions on the use of FMT in the setting of HSCT, neither for restoring the microbial flora after antibiotics or chemotherapy nor for the treatment of multidrug-resistant infections, let alone for the treatment of GvHD. Likewise, there are no guidelines on the use of probiotics or prebiotics either. Immunologic fragility and the risk of infectious complications and GvHD have not allowed FMT to be included in approved treatments in the HSCT setting. Only clinical trials are still ongoing in this setting, as reported in our paper. To date, the only recognized indication for FMT is in the second recurrence of clostridium difficile infection in the general population, as we have reported in the section 5.

Microbiota changes associated with GvHD

Comment: Line 345: Please include the explanation for the Simpson index.

Reply: We added the explanation of inverse Simpson index.

Comment: It would be appreciated to include the achievements about using pre- and probiotics in GvHD. The suggestion also refers to the section 5.

Reply: this is an invited review on the role of fecal microbiota transplantation. Therefore, we focused on it rather than pre- or probiotics. However, in section 3 we reported the more recent paper on the use of probiotics and prebiotics in the setting of GVHD. Although the administration of probiotics and prebiotics seems to be relatively safe, its use remains debated in this particular type of patient, and more studies are needed to confirm the potential benefit and safety of pre- and probiotics.

Fecal microbiota transplantation for GvHD

Comment: Line 401: Citing three publications does not support the information about many case reports.

Reply: we removed the term “Many”.

Technical suggestions

Comment:  Please consider the efforts to explain the abbreviation when you introduce it.

Reply: we revised abbreviations and added explanations where missing.

Comment: Additional caution is necessary to use the italic font for the names of microorganisms and the Latin terms.

Reply: we revised the paper according to the reviewer's suggestion.

Comment: Please replace the abbreviations bid and tid with the terms familiar to the broad readership.

Reply: We have replaced bid with “twice a day” and tid with “three times a day”.